# Oxidative Stress in Healthy and Pathological Red Blood Cells

**DOI:** 10.3390/biom13081262

**Published:** 2023-08-18

**Authors:** Florencia Orrico, Sandrine Laurance, Ana C. Lopez, Sophie D. Lefevre, Leonor Thomson, Matias N. Möller, Mariano A. Ostuni

**Affiliations:** 1Laboratorio de Fisicoquímica Biológica, Instituto de Química Biológica, Facultad de Ciencias, Universidad de la República, Montevideo 11400, Uruguay; forrico@fcien.edu.uy (F.O.); anclalop@fcien.edu.uy (A.C.L.); mmoller@fcien.edu.uy (M.N.M.); 2Laboratorio de Enzimología, Instituto de Química Biológica, Facultad de Ciencias, Universidad de la República, Montevideo 11400, Uruguay; lthomson@fcien.edu.uy; 3Centro de Investigaciones Biomédicas (CEINBIO), Universidad de la República, Montevideo 11800, Uruguay; 4Université Paris Cité and Université des Antilles, UMR_S1134, BIGR, Inserm, F-75014 Paris, France; sandrine.laurance@inserm.fr (S.L.); sophie.lefevre@inserm.fr (S.D.L.)

**Keywords:** erythrocyte, reactive oxygen species, antioxidant, oxidative stress, sickle cell disease, glucose 6-phosphate dehydrogenase deficiency, pyruvate kinase deficiency

## Abstract

Red cell diseases encompass a group of inherited or acquired erythrocyte disorders that affect the structure, function, or production of red blood cells (RBCs). These disorders can lead to various clinical manifestations, including anemia, hemolysis, inflammation, and impaired oxygen-carrying capacity. Oxidative stress, characterized by an imbalance between the production of reactive oxygen species (ROS) and the antioxidant defense mechanisms, plays a significant role in the pathophysiology of red cell diseases. In this review, we discuss the most relevant oxidant species involved in RBC damage, the enzymatic and low molecular weight antioxidant systems that protect RBCs against oxidative injury, and finally, the role of oxidative stress in different red cell diseases, including sickle cell disease, glucose 6-phosphate dehydrogenase deficiency, and pyruvate kinase deficiency, highlighting the underlying mechanisms leading to pathological RBC phenotypes.

## 1. Oxidative Stress in Healthy Red Blood Cells

Red blood cells (RBCs) are exposed to endogenous and exogenous oxidants, commonly referred to as reactive oxygen species (ROS) and reactive nitrogen species (RNS). These oxidants comprise a large group of molecules with different properties, including cellular sources and preferred molecular targets, which will be briefly discussed below, and have been discussed in more detail elsewhere [1,2].

One of the main mechanisms of endogenous oxidant production involves oxyhemoglobin (HbO_2_). The autoxidation of HbO_2_ occurs spontaneously at a low rate to yield superoxide (O_2_^•−^) and methemoglobin (Hb-F^III^, MetHb) [3]. Superoxide itself is a weak oxidant but can further react to make stronger oxidants. Superoxide can spontaneously dismutate to yield hydrogen peroxide (H_2_O_2_) and oxygen [4]. Hydrogen peroxide is a stronger oxidant that will react with thiols and metal centers [5]. Hydrogen peroxide reacts with HbO_2_ to yield ferryl hemoglobin, which can oxidize other proteins and lipids [6]. Furthermore, in the presence of one electron reductant such as Fe^II^, H_2_O_2_ can also generate hydroxyl radical (HO^•^), one of the strongest biological oxidants [4]. Hydroxyl radicals will react with most organic molecules at diffusion-controlled rates to yield organic radicals that can propagate oxidative damage [7].

RBCs will also be exposed to oxidants derived from endothelial and immune system cells, which generate nitric oxide (NO^•^), superoxide, peroxynitrite (ONOO^−^), H_2_O_2_, and hypochlorous acid (HOCl). Nitric oxide is produced by the endothelial enzyme nitric oxide synthase (NOS3) as a signal molecule to induce vasodilation and by immune system cells at larger amounts by inducible NOS2, that result in the formation of more potent oxidizing species that can kill invading pathogens [8].

Because of its small size and hydrophobicity, NO^•^ can diffuse virtually unhindered through cellular membranes [9]. In the RBCs, NO^•^ will react with HbO_2_ to give MetHb and nitrate and also with O_2_^•−^ to give the powerful oxidant ONOO^−^ [10,11]. Superoxide is also produced by NADPH oxidases in endothelial and immune system cells [12]. Although the ionic nature of superoxide limits its diffusion across cellular membranes, it has been observed to be transported by RBC band 3 bicarbonate/chloride exchange protein [13], and it can also protonate to the hydroperoxyl radical and diffuse across the lipid fraction of the membranes [14]. Peroxynitrite can use the same routes to diffuse into RBCs and cause intracellular oxidation, facilitated transport of the anion by band 3, and simple diffusion of the neutral peroxynitrous acid [15]. Peroxynitrite per se is a potent oxidizing molecule, but it rapidly reacts with CO_2_ in solution to give the more potent oxidants nitrogen dioxide (NO_2_^•^) and carbonate radical (CO_3_^•−^) [16]. Nitrogen dioxide diffuses very rapidly across cellular membranes and reacts mainly with intracellular thiols but also with lipids in the membrane [1,17]. Carbonate radicals cannot diffuse across cellular membranes and will react preferentially with proteins [18].

Hydrogen peroxide produced by endothelial and immune-system cells directly or indirectly by NADPH oxidases also acts as a signaling molecule in the vascular system modulating endothelial cell proliferation and survival, for instance [19]. This exogenous H_2_O_2_ can rapidly diffuse across the RBC membrane to oxidize intracellular targets, through an aquaporin-independent pathway, likely by simple diffusion [2]. Hypochlorous acid is produced enzymatically by myeloperoxidase that is released by neutrophils and monocytes upon infection-related or inflammatory stimuli, using chloride and H_2_O_2_ as substrates [20]. Hypochlorous acid freely diffuses across the RBC membrane and oxidizes intracellular targets [21].

All these oxidants produced in and around the RBC can oxidize biomolecules in the RBC, but different oxidants have different preferential targets and lead to different types of damage, as will be discussed below.

### 1.1. Main Targets of Oxidants in RBCs

As mentioned above, both proteins and lipids can be damaged by oxidants in RBCs. The ultimate result of oxidative damage to RBCs is hemolysis, the loss of membrane integrity, and the release of hemoglobin and other intracellular proteins. Free hemoglobin is particularly toxic, and this is evident in several RBCs diseases [22].

The membrane of RBCs is composed of phospholipids, cholesterol, glycolipids, and proteins (some of them glycosylated). The polyunsaturated fatty acids (PUFA), 18% of the total fatty acids in RBCs [23], are the lipid components that are more susceptible to oxidation in a series of reactions that trigger lipoperoxidation. The first event is the abstraction of bis-allylic hydrogen from a polyunsaturated fatty acid to yield a lipid-derived radical, which rapidly reacts with molecular oxygen to yield a lipid-derived peroxyl radical (LOO^•^). This LOO^•^ can subsequently subtract hydrogen from a neighboring PUFA, and the reaction propagates as a chain reaction [24]. The further oxidation of LOOH yields reactive aldehydes such as hydroxynonenal and malondialdehyde. The lipid peroxidation propagation can be stopped by lipid-soluble antioxidants, such as α-tocopherol [24]. Also, a minor fraction of phospholipids (10–15%) is present as plasmalogens, which present a vinyl ether hydrocarbon group that has been associated with lipid antioxidant capacity in vitro [25,26].

Fresh normal RBCs do not contain products of lipid peroxidation, but increased membrane lipid peroxidation is evident in many RBCs diseases, such as thalassemia, unstable hemoglobin disease, and sickle cell disease [27]. RBC diseases are also accompanied by an increased susceptibility to lipid peroxidation. Additionally, lipid peroxidation products also increase during the storage of lipids for transfusion, especially when no leukoreduction is performed [28]. In many of these cases, the oxidation of lipids is associated with the oxidation of protein and crosslinking to cytoskeleton and membrane proteins [29]. Notably, much of the lipid oxidation is catalyzed by HbO_2_ and is inhibited when hemoglobin is present as MetHb [27]. A possible explanation is that monomeric or unstable hemoglobin is associated with the RBC lipid membrane and catalyzes the formation of oxidizing radicals, such as HO^•^, in situ, that cause lipid oxidation and protein cross-linking. In this line, hydrophobic oxidants like cumene hydroperoxide or tert-butyl hydroperoxide, which partition favorably in the membrane of RBCs, have been consistently observed to cause more damage than water-soluble H_2_O_2_ [30,31,32]. Recently, increased levels of the six-transmembrane epithelial antigen of prostate 3 (Steap3) protein in mice has been associated with increased lipid oxidation in RBCs and hemolysis [33]. The protein Steap3 reduces Fe^3+^ to Fe^2+^ after DMT1 transmembrane transport, and its deletion in mice causes iron deficiency anemia [34]. The mechanism of cellular damage is proposed to involve the reduction of free iron to Fe^2+^, followed by a Fenton-type reaction with H_2_O_2_ to produce HO^•^, that then oxidizes lipids and proteins [33].

The oxidation of hemoglobin is most easily recognized and has been found to be involved in many of the oxidative damage to RBCs. Some drugs, such as phenylhydrazine, can cause hemolytic anemia by means of hemoglobin oxidation that leads to the formation of hemichrome, a misfolded form of hemoglobin, its precipitation to form Heinz bodies, visible by microscopy, and then to hemolysis [35]. Similar effects have been observed in RBCs from mice deficient in peroxiredoxin2 (Prx2), an important antioxidant enzyme discussed below [36]. The oxidation of the cytoskeleton has also been observed during the storage of RBC for transfusion, associated with an increase in protein carbonyls and crosslinking [37]. Atomic force microscopy showed that spectrin filaments are altered during storage, forming thicker fibers and loss of connections [38].

### 1.2. Antioxidant Cellular Mechanisms in RBCs

Although the RBCs are exposed to large amounts of oxidants, both from endogenous and exogenous sources, they are well prepared to resist. Robust antioxidant defenses allow normal RBCs to survive 120 days in circulation. The main defenses against oxidant damage are provided by different enzymatic systems, aided by low molecular weight antioxidant and electron-rich molecules. The antioxidant defenses ultimately rely on the reducing power of NADPH, obtained from the oxidation of glucose by the pentose phosphate pathway.

Glucose is transported across the RBC membrane by the highly abundant GLUT1 transporter driven by the concentration gradient. In the cytosol, glucose is phosphorylated to glucose-6-phosphate (G6P) by hexokinase. A large fraction of glucose is used in glycolysis to produce the ATP necessary to keep important cellular functions working, such as the Na^+^/K^+^ ATPase and Ca^2+^ ATPase pumps, and NADH that is mostly used to reduce pyruvate to lactate and is also used to reduce MetHb to HbO_2_ by methemoglobin reductase. A smaller fraction of the glucose is used in the pentose phosphate pathway to produce NADPH, which will provide the reducing power necessary to keep glutathione (GSH) and the thiol-dependent antioxidant enzymes reduced. NADPH is produced in the steps catalyzed by glucose-6-phosphate dehydrogenase (G6PDH) and 6-phosphogluconate dehydrogenase. The deficiency in G6PDH activity has important consequences in the capacity of RBCs to deal with oxidative stress and is recognized as a health problem that will be dealt with below.

Low molecular weight antioxidants include α-tocopherol in the plasma membrane to prevent lipid peroxidation, ascorbate, and GSH in the cytosol that can rapidly react with NO_2_^•^ or HOCl. Reduced ascorbate can reduce tocopheroxyl radicals in the membrane, and oxidized ascorbate can be reduced with GSH, forming a network of antioxidant power. It appears that the most important roles of low molecular weight antioxidants are to protect the lipids from oxidation, repair molecular radicals, and act as an additional reducing power reservoir [2].

RBCs also contain a large network of antioxidant proteins that can react and deactivate oxidant species very rapidly, preventing the damage of protein and lipids essential for RBC function. Some of them need the assistance of other proteins to function, some do not, but most of them act in a concerted fashion to decrease to a minimum the chances of damage to important molecules in RBCs. Given their relevance, a detailed discussion of the most important antioxidant proteins is given below.

Superoxide dismutase

Superoxide dismutases (SODs) are antioxidant enzymes that catalyze the disproportionation of superoxide radicals and, therefore, constitute the first barrier against oxidative damage in cells. These enzymes contain metal atoms in their structure, based on which they can be classified into four groups, FeSOD, Cu, ZnSOD, MnSOD, and NiSOD [39]. Cu, ZnSOD (SOD1) is the isoform found in RBCs. It is present at an approximate concentration of 2 µM [40] and is associated to 95% of the entire pool of Cu^2+^ in the cell [41].

SOD1 is organized as a dimer, with identical active sites in each subunit. The Cu^2+^ atom is the redox-active metal that is sequentially reduced and oxidized by two superoxide molecules to yield one molecule of oxygen and one of H_2_O_2_ [42,43]. Zn^2+^, on the other hand, is involved in structural stabilization and ensures correct coordination and reducing the potential of Cu^2+^ [44]. This process is very efficient, as the enzyme manages to increase the rate of the spontaneous reaction by four orders of magnitude, reaching practically diffusion-controlled values [43].

Kinetic model calculations estimate that if SOD1 were not present in RBCs, superoxide would increase its levels to values 100 times higher than the basal concentrations [3]. Experiments performed with SOD1 deficient mice show the consequences are a rise in MetHb and lipid peroxidation levels, accumulation of oxidized carbonic anhydrase, and, lastly, anemia due to a decrease in lifespan and oxidative damage of RBCs [45,46].

SOD1 is characterized by having disordered regions with undefined secondary or tertiary structures that can change rapidly in conformation. These regions appear to be often post-translationally modified and are proposed to have regulatory roles [47,48]. Furthermore, SOD1 was reported to be inactivated and more susceptible to proteolysis upon reaction with its product H_2_O_2_ [49,50]. However, given the abundance and reactivity of RBC peroxidases, this process is not likely to be of relevance in vivo. Other alterations in SOD1 activity are reported in circumstances that are often linked to oxidative stress, such as Alzheimer’s disease and aging [51,52]. Because of this, it is often considered an interesting marker to study regarding oxidative stress.

Glutathione peroxidase

Glutathione peroxidases (GPxs) are a family of enzymes that catalyze the reduction of hydroperoxides with varying substrate and electron donor specificities. The most abundant isoform in RBCs is GPx1, with an estimated concentration of 1.5 µM [53]. GPx1 is a homotetrameric enzyme with a selenium-containing active site [54]. Through its catalytic cycle, the enzyme is oxidized in order to reduce the peroxide and is later regenerated by GSH. The result is the formation of glutathione disulfide (GSSG), which in turn is reduced by glutathione reductase (GR) and NADPH [55].

Since its discovery, GPx1 has been associated with the protection of RBCs from oxidative damage, particularly inflicted by endogenously produced oxidants present in low concentrations [55,56]. It is proposed to act as a second barrier to H_2_O_2_, below Prx2, and to be especially implicated in the detoxification of small organic peroxides that could generate damage to the lipid membrane [3,53,57].

A second glutathione peroxidase with a similar role, GPx4, can also be found in RBCs, albeit 20 times less abundantly than GPx1 [58]. It is also a selenoenzyme, but it clearly differs in electron donor specificity, being able to be reduced by other thiols [59]. Furthermore, as it is a monomeric enzyme, it is allowed to metabolize more complex substrates, such as cholesterol and phospholipid hydroperoxides, even when they are bound to the membrane [59,60,61].

Although not frequent, deficiencies in GPx have been reported and subject the patients to a clinical profile similar to the observed G6PDH deficiency. Mainly, drug-induced hemolysis and hemoglobinuria have been described [62,63].

Glutathione reductase

GR is a flavoprotein that belongs to the pyridine nucleotide-disulfide oxidoreductase family of enzymes. Structurally, it is a dimeric protein with specific binding sites to FAD, NADPH, and GSSG. Both these subunits are involved in the mechanism of catalysis, where a redox-active disulfide in the active site of the enzyme catalyzes the conversion of GSSG into two molecules of GSH. To start another cycle, this disulfide is later reduced by NADPH [64,65].

In physiological conditions, the concentration of oxidized GR is estimated to be very low since it is continuously being reduced [65,66]. GR is thus a very important part of the antioxidant system in RBCs, as it collaborates in maintaining GSH levels more than ten times higher than those of GSSG [67], in addition to helping in the regulation of the NADP^+^/NADPH pool.

Given its role, the absence or decrease in GR activity entails various physiological complications. The causes behind such abnormalities vary and can be related or not to FAD metabolism, for example, an insufficient dietary intake of riboflavin or an inability to obtain FAD from riboflavin [68]. It has also been reported the existence of patients with inherited mutant variants of GR, where the altered enzyme inefficiently binds FAD or is eliminated due to problems in folding [69]. In these scenarios, as the capacity of RBCs to balance oxidative stress is impaired, the consequences are shown to be favism and drug-induced hemolysis, an increase in osmotic fragility, and a reduction of lifespan in RBCs, as well as spherocytosis [70,71].

Peroxiredoxin 2

Peroxiredoxins (Prxs) are widely extended antioxidant proteins with thiol-dependent peroxidatic activity. Three different members of this family have been found in RBCs, namely Prx 1, Prx2, and Prx6 [72], although Prx2 is by far the most abundant. In fact, it represents the third most abundant protein in the RBCs, ranging between 240 and 410 µM [73,74].

Prx2 can metabolize peroxides of different natures, such as lipid hydroperoxides, peroxynitrite, and especially H_2_O_2_, with which it reacts at a very high rate [72,75]. Since it belongs to the typical 2-Cys type of peroxiredoxins, Prx2 forms homodimers with identical active sites in each subunit during catalysis. The catalytic cycle starts with the oxidation of one monomer peroxidatic cysteine to cysteine sulfenic acid upon reaction with the peroxide, followed by the formation of an intermolecular disulfide bond between this residue and a resolutive cysteine of a second monomer. The enzyme is later regenerated by thioredoxin (Trx), thioredoxin reductase (TR), and NADPH [75,76]. During this process, Prx2 can be hyperoxidized to cysteine sulfinic acid in its peroxidatic cysteine, which can only be reduced enzymatically by sulfiredoxin or further oxidized to an irreversible sulfonic form [77,78]. Probably regulated by its redox state, a dynamic equilibrium is established in vivo where the enzyme can exist as a dimer when oxidized or as a decamer when reduced or hyperoxidized. In addition, these decamers were shown to interact with each other and form greater molecular weight structures [76,79,80,81].

Due to its high abundance and reaction rates, Prx2 has been proposed to act as a first line of defense against endogenous H_2_O_2_ produced in RBCs, mainly by the autoxidation of hemoglobin [3,53,82]. Because of this, it is often considered a sensitive marker to oxidative stress. It can also act as a chaperone, binding to hemoglobin and preventing its aggregation due to oxidative damage [83,84,85]. In fact, studies report that deficiencies in Prx2 cause an increase in H_2_O_2_ and MetHb levels, alterations in cell morphology, formation of Heinz bodies, and hemolytic anemia [36,57]. Furthermore, Prx2 is also able to bind the cytosolic domain of band 3, so it could play a role in defending band 3 and other proteins in the membrane, as well as the lipidic fraction, from oxidants [85,86].

Apart from its peroxidase role, Prx2 has been previously associated with potassium transport in RBCs. The increase in calcium levels promotes the translocation of Prx2 to the membrane, where it can interact with the Gardos channel and thus alter potassium efflux and cell volume. However, the causes and mechanisms are not fully understood yet to this day [73,81]. More recently, a signaling role has also been proposed for peroxiredoxins, given their high reactivity with H_2_O_2_ and the ability to interact with other proteins. Post-translational modifications observed in Prx2, such as phosphorylation, proteolysis, and even the hyperoxidation of its peroxidatic cysteine, are involved in the regulation of its peroxidase activity and could affect its participation in redox signaling pathways [87,88,89].

Thioredoxin

Thioredoxins (Trxs) are small antioxidant oxidoreductases. They can reduce a wide variety of protein disulfides since the active site surface of these enzymes can undergo chaperone-like conformational changes to accommodate different proteins [90]. Some targets include methionine sulfoxide reductases, sulfonucleotide reductases, and transcription factors, such as Ref-1 [91,92]. In RBCs, cytosolic Trx1 is particularly relevant for regenerating Prx2 and maintaining a reduced environment within the cell [75].

The catalytic mechanism of Trx1 depends on two cysteine residues to reduce its substrates. At first, a thiolate in Trx1 attacks the intermolecular disulfide bond that unites Prx2 dimers, forming a mixed disulfide. This disulfide is then disrupted by a second cysteine residue, obtaining a reduced Prx2 and an oxidized Trx1 that can be reduced by TR and NADPH [90]. There are three extra cysteine residues in the Trx1 structure, reported to be easily oxidized and implicated in the formation of molecular aggregates or oligomers and inactivation of the enzyme. These are suspected to have a physiological, possibly regulatory role [93,94,95,96].

Thioredoxin reductase

TR, like GR, is a flavoprotein from the pyridine nucleotide-disulfide oxidoreductase family. Both these enzymes are highly similar in structure, as TR also acts as a homodimer with FAD and NADPH binding domains, as well as an interface domain [91,97]. However, they differ in their active sites because TR is a selenoprotein [98]. During the catalytic cycle, a selenenylsulfide bond is established between TR dimers after the reduction of the oxidized thiols in the substrate, which can be later reduced by a chained electron transfer from NADPH and FAD [97,99,100].

TR has been shown to have multiple possible substrates, among which are glutaredoxins and thioredoxins, hence collaborating with the antioxidant proteins in the RBCs. Furthermore, it can reduce smaller molecules that could react with H_2_O_2_ directly and is even proposed as an alternative system to reduce GSH in other organisms, so this could possibly occur in mammals as well [96,101,102]. There is also evidence that TR could play a role in the enzyme-mediated dehydroascorbic acid reduction in RBCs [103].

Glutaredoxins

Glutaredoxins (Grxs) are a family of small and ubiquitous thiol-disulfide oxidoreductases. They have a considerable role in maintaining the sulfhydryl homeostasis in cells by reducing inter and intramolecular disulfides in proteins, as well as mixed protein-glutathione disulfides. Classified into two groups, Grxs can present either a dithiolic mechanism of catalysis or a monothiolic one. In the first case, two cysteine residues are involved, attacking the disulfide in the target protein and prompting its reduction while the enzyme gets oxidized. Grx is later regenerated by glutathione, with the formation of a glutathionylated intermediate. In the monothiol mechanism, the targets are glutathionylated proteins, and only one cysteine is needed for its reduction [104,105,106].

Red blood cells contain two isoforms of Grx, specifically Grx1 and Grx3. However, the dithiolic Grx1 is the most abundant, with reported concentrations of 4–8 µM [58]. It has been proven that Grx1 is capable of reducing oxidized forms of hemoglobin and membrane proteins, as well as regenerating metabolic enzymes phosphofructokinase, pyruvate kinase, and glyceraldehyde-3-phosphate dehydrogenase that are susceptible to inactivation by oxidative damage [107,108,109]. Nevertheless, glutathionylation of proteins is not exclusive to oxidative stress conditions, as it is also proposed to have a role in signaling pathways. In fact, Prx2, a candidate for the transduction of redox signals, can be glutathionylated in one or both of its active site cysteines and is deglutathionylated by Grx1 [110].

The role of Grx3 is not clear in mature RBCs. However, it could be particularly relevant during erythropoiesis since monothiolic Grxs, as is Grx3, are involved in iron homeostasis and the assembly of Fe-S clusters [111]. In other organisms, depletion of Grx3 during embryonic development affects the maturation of hemoglobin [112]. Grx5, also a monothiolic Grx, was shown to affect heme synthesis in erythroblasts, resulting in sideroblastic anemia in humans [113].

Catalase

Catalases are enzymes that catalyze the dismutation of H_2_O_2_ into water and oxygen, thus protecting cells from oxidative damage. Human catalase organizes as a tetramer. It contains four heme groups and binds four molecules of NADPH, one in each subunit [114,115]. The tetramerization is essential for its activity, as it is proposed to allow the correct cycling of the enzyme and keep the iron of heme groups away from the protein surface, therefore avoiding the formation of hydroxyl radicals [114].

During catalysis, the elimination of H_2_O_2_ occurs in two sequential steps. First, ferric heme reduces one molecule of H_2_O_2_ to water, resulting in the formation of an oxyferryl species named compound I. In the second step, compound I is reduced using another molecule of peroxide, yielding water and oxygen and returning catalase to its basal redox state [114,116]. In a series of reactions involving superoxide and H_2_O_2_, compound I can be transformed into inactive forms of the enzyme (compounds II and III) [117]. NADPH, which stays in a reduced state when bound to catalase, is proposed to be there to prevent this inactivation by reacting with an intermediate species and blocking compound II accumulation [118].

RBCs are among the cells with the highest catalase activity in the organism, with an estimated subunit concentration of 11 µM [53]. Given the abundance and high reaction rates of Prx2, catalase is not very relevant in metabolizing endogenous peroxide in these cells [53]. However, it differs from other enzymes as it cannot be saturated with H_2_O_2_ [119]. This comes into play in circumstances where H_2_O_2_ levels are sufficiently high to react completely with all Prx2 and GPx available and deplete cellular NADPH, which is only slowly recovered [53]. This is often the case when experiments are performed in vitro with a relatively high concentration of H_2_O_2_ and low hematocrit, and has often led to believe that catalase is the main enzyme in the detoxification of exogenous H_2_O_2_ [53]. In line with this, experiments performed with mice RBCs deficient in catalase show they are more sensitive to high H_2_O_2_ concentrations, suffering an increase in hemoglobin oxidation [57].

Considerations on the antioxidant systems in RBCs

RBCs are exposed to oxidants from endogenous sources and exogenous sources that can react with RBC lipids and proteins, leading to molecular damage and eventually to membrane rupture and hemolysis. To prevent these harmful effects, RBCs are well prepared with enzymatic and low molecular weight antioxidants (Figure 1). Actually, the enzymes Prx2, Gpx, and catalase appear to deal with most of the reactive oxygen and nitrogen species attacking RBCs. Low molecular weight antioxidants appear to be more important in preventing lipid oxidation (α-tocopherol/ascorbate/GSH system) and acting as a reducing power reservoir. At the same time, both antioxidant systems ultimately rely on the reducing power provided by NADPH from glucose in the pentose phosphate pathway.

Genome-wide association analysis has identified candidate genes that modify RBC susceptibility to osmotic hemolysis, including spectrin α-chain, ankyrin 1, aquaporin 1, and band 3, most of them involved with cytoskeleton stabilization or osmosis-driven water transport [120]. Other genes identified included hexokinase 1, the first enzyme in glucose metabolism, stress kinase MAPKAPK5, and the mechanosensitive calcium channel PIEZO1 [120]. Oxidative hemolysis is associated with changes in G6PDH, Grx, and GPx4, important proteins in redox metabolism and repair, and SEC14-like 4, a protein related to the transport of phospholipids and α-tocopherol [120]. Resistance to oxidative hemolysis was also associated with higher levels of GSH, a higher activity of the pentose phosphate pathway, and improved protein damage repair mechanisms [121].

## 2. Oxidative Stress in Red Blood Cell Diseases

As developed above, RBCs’ oxidative stress strongly depends on the balance among pathophysiological mechanisms producing ROS and enzymatic and non-enzymatic antioxidant systems. In this second part of the review, we aim to explore several RBC diseases where this balance is strongly altered both by increased ROS production or by diminished antioxidant capacity. The selection of RBC diseases is not exhaustive, but representative of those involving different oxidative stress causes.

### 2.1. Sickle Cell Disease

Sickle cell disease (SCD) is an inherited genetic disorder resulting in the production of an abnormal hemoglobin S (HbS) that undergoes deoxygenation-dependent polymerization [122,123]. The repeated cycles of HbS polymerization induce RBCs’ shape distortion, cell rigidity, cell membrane alteration, and fragility, ultimately resulting in intravascular and extravascular hemolysis. Behind this primary described pathological mechanism, SCD pathophysiology appears to be more complex and involves an intricate network of molecular and cellular partners. In fact, in addition to HbS polymerization, an imbalance of the redox status is also observed in SCD due to an increase in the production of ROS and/or RNS conjugated to an impairment of the antioxidant systems (Figure 2 SCD panel). For example, peroxynitrite is involved in the oxidation and nitration of several intracellular targets (thiols, protein-membrane, lipids), leading to breakage of DNA, impairment of cell signaling and cell death (reviewed in [124,125]). In SCD, oxidative stress can arise from sickle RBCs and/or activated neutrophiles, platelets, and endothelial cells (ECs). Several erythroid and non-erythroid mechanisms have been described accounting for this pro-oxidant environment: (i) HbS autooxidation, (ii) heme and iron release, (iii) increased NADPH oxidase and endothelial xanthine oxidase (XO) activity, (iv) decreased NO^•^ bioavailability, (v) erythroid mitochondrial retention [126].

HbS autoxidation

HbS is very unstable and could easily undergo autoxidation in the presence of oxygen. The reaction leads to the production of MetHb that no longer binds oxygen and O_2_^•−^ that dismutates to H_2_O_2_ [127,128,129,130]. This results in oxidative damage of the RBC membrane and lipid and protein oxidation, leading to hemolysis and release of toxic heme leading to the exacerbation of the pro-oxidative environment (see below). HbS-induced oxidative stress leads to post-translational modifications of hemoglobin (notably oxidation of Cys93 and ubiquitination of Lys96 and Lys145 of the β globin), phosphorylation of band 3, the most abundant protein of the RBC membrane, and ubiquitination of other erythroid proteins. ROS-induced band 3 modification induces its clusterization and dissociation from membrane/cytoskeleton complexes, leading to RBC membrane disorganization and potentially microparticle formation [131,132,133]. Several studies have highlighted the role of microparticles in several SCD complications, such as vaso-occlusion and kidney dysfunction [134,135,136].

Hemolysis: heme and iron release

Repeated cycles of sickling/unsickling lead to the fragilization of the RBC membrane and thus to hemolysis that results in the release of extracellular hemoglobin, free heme, and free iron, all highly toxic to the vasculature by triggering vascular oxidative burden [22,137,138]. In fact, oxidative stress generated at the erythroid levels can affect not only RBCs but also neutrophils, monocytes, and endothelial cells. The released heme and ATP from hemolyzed RBCs will act as damage-associated molecular patterns (DAMPs), promoting the activation of endothelial cells, macrophages, and neutrophils through different cellular pathways involving several receptors such as P2X7, toll-like receptor 4 (TLR4) or other unidentified receptors. Those activation processes trigger the expression of adhesion molecules at the cell surface and also pro-inflammatory mediators resulting in the exacerbation of the pro-inflammatory and oxidant environment. These can ultimately lead to vaso-occlusion and other SCD complications. Heme promotes adhesion events and thus vaso-occlusion through the von Willebrand factor (vWF) release from endothelial granules, inter cellular adhesion molecule (ICAM-1), vascular cell adhesion molecule (VCAM-1), and P-selectin expression at the surface of the vessel wall. Thus, it promotes leukocyte recruitment to the vessel wall and leukocyte/sickle RBC interactions. In the SCD context, heme has also been shown to promote neutrophil activation and Neutrophil Extracellular Traps (NETs) that are composed of decondensed chromatin with cytoplasmic protein [139,140,141]. Those NETs can, in turn, contribute to the activation of the vascular system through the activation of the TLR4/TLR9 signaling pathways, thus exacerbating the oxidative environment.

NADPH oxidase and XO activity

NADPH oxidase is one of the major enzymes responsible for the production of O_2_^•−^ in leukocytes, RBCs, and endothelial cells. ROS produced by erythroid NADPH contributes to the exacerbation of erythroid dysfunction by exacerbating cell stiffness resulting in the increase of hemolysis [142]. Xanthine oxidase (XO) is also responsible for a large part of the production of O_2_^•−^ and H_2_O_2_. Its activity is increased in SCD plasma, but its source remains to be clearly identified [143].

NO^•^ bioavailability

NO^•^ plays an important role in vascular homeostasis and physiology. Notably, it acts on smooth muscle cells by regulating the vascular tone as a vasodilator, and on endothelial cells through downregulation of the expression of members of the selectin family, such as ICAM-1 and VCAM-1 [144]. NO^•^ could also inhibit platelet activation [145]. Interestingly, NO^•^ may also act on the RBC itself by modulating its deformability through, in part, the soluble guanylate cyclase (sGC) [146,147].

In SCD, hemolysis and consecutive free extracellular hemoglobin release lead to NO^•^ scavenging, thus decreasing its bioavailability in the circulation. Furthermore, the production and release of O_2_^•−^ may participate in the decrease in NO^•^ through its reaction to form ONOO^−^ [148]. Therefore, the decreased NO^•^ bioavailability in SCD, negatively affects vascular tone regulation and expression of adhesion proteins.

Erythroid mitochondrial retention

Recently, Jagadeeswaran et al. showed that RBCs from SCD patients retain mitochondria [149], and this was confirmed by other groups [150,151,152]. However, the functionality of these mitochondria remains controversial. Some studies showed that they were still functional and that the mitochondrial retention was associated with high levels of ROS, but some of these observations have been made in SCD mice model or in a population of erythroid circulating cells that might also include reticulocytes, i.e., immature RBCs [149,151,152]. Another group did not detect any activity of these retained mitochondria in mature RBCs [150]. A clear link between mitochondrial retention and the increased oxidative stress in SCD remains to be fully determined as well as the mechanism leading to this mitochondrial retention.

It is well established now that oxidative stress plays an essential role in SCD pathophysiology and in complication occurrence. However, it appears that oxidative mechanisms are considerably complex as they involve not only the RBC as the primary pathological cell target but also vascular endothelial cells, monocytes, and neutrophils. The complexity is heightened by the intimate interplay between oxidative mechanisms and inflammation with the activation of innate immune cells and the production of pro-inflammatory mediators. A vicious circle sets in, exacerbating the pro-oxidative, pro-inflammatory, pro-coagulant, pro-adherent environment. This highly toxic milieu is deleterious in the short-term with the appearance of acute complications and also deleterious in the long-term, with end-organ damage. Consequently, new drugs targeting oxidative stress have been developed to counteract its detrimental consequences. To date, the main antioxidant therapy that has shown some benefits in SCD clinical trials is L-glutamine, an amino acid needed for the synthesis of nucleotides as NAD. Supplementing with L-glutamine could reduce the erythroid oxidative process and protect RBCs from oxidative damage. However, this treatment has shown limitations as some SCD patients did not tolerate the treatment, and it seems to fail to counteract anemia and hemolysis [153,154]. This observation means that the mechanism underlying pro-oxidative stress in SCD requires a lot more investigation in order to identify new potential therapeutical targets.

### 2.2. Glucose 6-Phosphate Dehydrogenase Deficiency

G6PDH catalyzes the first reaction in the pentose–phosphate pathway, oxidizing glucose-6-phosphate to 6-phosphogluconate and reducing NADP to NADPH, which is essential to provide reducing equivalents to several antioxidant systems [155,156,157], as discussed above.

The G6PDH deficiency (G6PD) is a chromosome X-linked highly polymorphic genetic disorder characterized by the reduced activity of the enzyme. Although most G6PD patients do not normally present clinical manifestations, RBCs from these patients present lower levels of NADPH and are more susceptible to oxidative stress (Figure 2 G6PD panel) induced by the action of drugs, anesthetics, infections, and metabolic disturbances [157,158,159,160], leading to hemolytic anemia and various health complications (reviewed in [156,157]).

G6PD is usually associated with favism, a hemolytic anemia syndrome induced by the ingestion of fava beans [161,162]. However, even though patients presenting fava bean intolerance carry some G6PD polymorphism, not all G6PD patients are intolerant to fava beans. Actually, different metabolites from fava beans, such as vicine and divicine, are highly oxidant and could induce hemolysis by depleting the antioxidant capacity of RBCs in a mechanism similar to that of synthetic drugs [163,164,165].

Interestingly, some of the drugs and compounds inducing hemolytic anemia in G6PD patients are not able to induce RBC hemolysis in vitro [163,164,165], supporting the hypothesis of other genetic factors contributing to the hemolytic phenotype [166]. Recently, Dinarelli et al. reported that RBCs from G6PD patients stored for 6–12 days were surprisingly less sensitive to hemolysis. Authors suggest that these aged RBCs presented a metabolic regulation leading to lower energy consumption and higher stress resistance [167]; however, this hypothesis should be further confirmed by studies including a higher number of patients. Moreover, these results are contradictory with those of Francis et al., which show that, after 42 days of storage, the quality of post-transfusion RBCs is significantly lower in G6PD patients compared with control subjects [168]. Infections, both from bacterial [169,170] or viral [171,172,173] origin, are also able to induce hemolytic anemia in G6PD patients, probably by inducing ROS production by circulating phagocytes.

Moreover, the severity of the clinical phenotype is patient-dependent. Looking for susceptibility factors, Tang et al. studied the metabolome changes in control or G6PD patients challenged by diamide-induced ROS production. They reported that diamide induced significant changes in RBC from G6PD patients leading to severe and irreversible loss of deformability [174].

Finally, as G6PD alters several cellular processes under oxidative stress and is frequently associated with anemia, it could be expected a deficient RBC maturation. However, in vitro differentiation of CD34^+^ hematopoietic progenitor cells isolated from patients with different G6PD severity did not show any alteration in progenitor proliferation, nor differentiation or enucleation [175].

### 2.3. Pyruvate Kinase Deficiency

Pyruvate kinase (PK) is a critical enzyme in the glycolytic pathway, catalyzing the conversion of phosphoenolpyruvate (PEP) to pyruvate and generating ATP [176]. Pyruvate kinase deficiency (PKD) is an autosomal (chromosome 1q21) recessive genetic disorder that affects RBCs’ ability to generate energy, leading to various degrees of hemolytic anemia [177,178,179,180].

Lacking mitochondria, mature RBCs’ ATP production depends exclusively on glycolysis. Thus, impaired or reduced PK activity in PKD patients leads to a dramatic decrease in ATP levels, which are necessary for maintaining the cell’s integrity and deformability [181,182]. Indeed, the main RBC membrane pumps controlling calcium, sodium, and potassium transport across the RBC membrane are P-type ATPase pumps, whose activity depends on ATP concentration. Reduced activity of these pumps leads to an altered ion balance that then induces water leak leading to RBC dehydration [183] (Figure 2 PKD panel). As a consequence, RBCs from PKD patients present altered membrane properties and become more susceptible to premature destruction at the spleen, leading to hemolytic anemia [184]. Other than the elimination of altered mature RBCs, PKD patients also present a diminished number of reticulocytes, which are most susceptible to low ATP levels.

Another consequence of PK deficiency is the accumulation of glycolytic intermediates such as 2,3-biphosphoglycerate (2,3-BPG), which diminishes the O_2_-hemoglobin affinity favoring the tissue oxygenation that could partially compensate for anemia [185,186].

In a recent elegant article, Roy et al. developed a metabolomic approach to characterize some changes in metabolism pathways from PKD patients. They demonstrated that RBCs from PKD patients present higher levels of oxidative stress markers, such as polyamines, sulfur-containing compounds, and deaminated purines, correlated with increased pentose phosphate pathways metabolites. Moreover, these patients also showed higher levels of poly- and highly-unsaturated fatty acids and acyl carnitine [187].

## 3. Conclusions

During evolution, mammals’ RBCs have lost nuclei, mitochondria, and other organelles, improving their efficiency in transporting and distributing oxygen and favoring the emergence of animal species with high energy demands.

However, the dark side of this process is the lack of transcriptional and translational tools to regulate oxidative stress. Indeed, RBCs need to “resist” during 120 days lifespan to the oxidative challenge. Oxidants derive both from endogenous (HbO_2_ autoxidation) and exogenous sources (endothelial and white blood cells). Normal RBCs are prepared to deal with these oxidants with an efficient enzymatic and low molecular weight antioxidant system that acts in concert. These antioxidant defenses of the RBCs depend ultimately on the reducing power of NADPH, obtained in the pentose phosphate pathway.

Several RBC diseases are associated with exacerbated oxidative stress, either presenting increased production of ROS such as SCD or diminished antioxidant capacity (G6PD and PKD) (Figure 2). All of them are genetic diseases presenting a wide and patient-dependent spectrum of clinical manifestations. Another shared characteristic is that other than promising genetic therapy, there are still no curative treatments for these diseases. As patient susceptibility is probably related to individual differences in genetic and metabolic landscape, the upcoming challenge could be to take advantage of the highly improved genomic, proteomic, and metabolomic approaches to identify possible targets useful for personalized treatments.

## Figures and Tables

**Figure 1 biomolecules-13-01262-f001:**
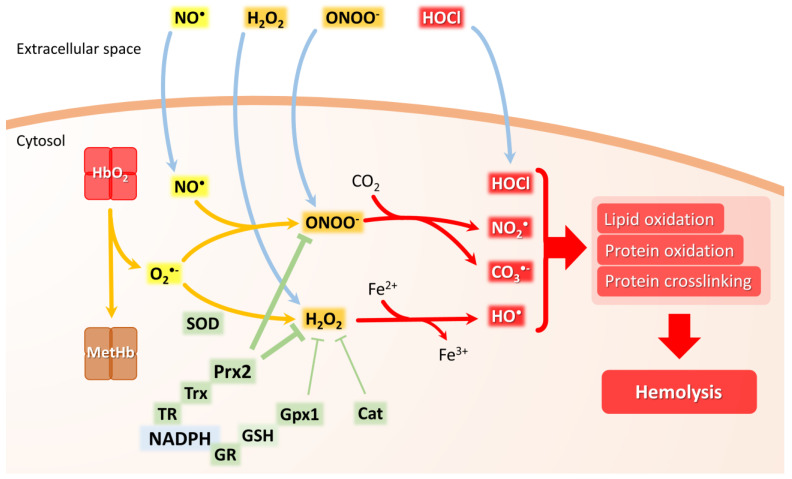
Scheme summarizing main endogenous and exogenous sources of ROS and RNS in the RBCs as well as the principal antioxidant actors. HbO_2_: Oxyhemoglobin; MetHb: methemoglobin; O_2_^•−^: superoxide anion; NO^•^: nitric oxide; H_2_O_2_: hydrogen peroxyde; ONOO-: peroxynitrite; HO^•^: hydroxyl radical; HOCl: hypochlorous acid; CO_3_^•−^: carbonate radical; NO_2_^•^: nitrogen dioxide radical; superoxide dismutase: SOD; catalase: Cat; glutathione reductase: GR; reduced glutathione: GSH; glutathione peroxidase 1: Gpx1; thioredoxin reductase: TR; thioredoxin: Trx; Peroxiredoxin 2: Prx2.

**Figure 2 biomolecules-13-01262-f002:**
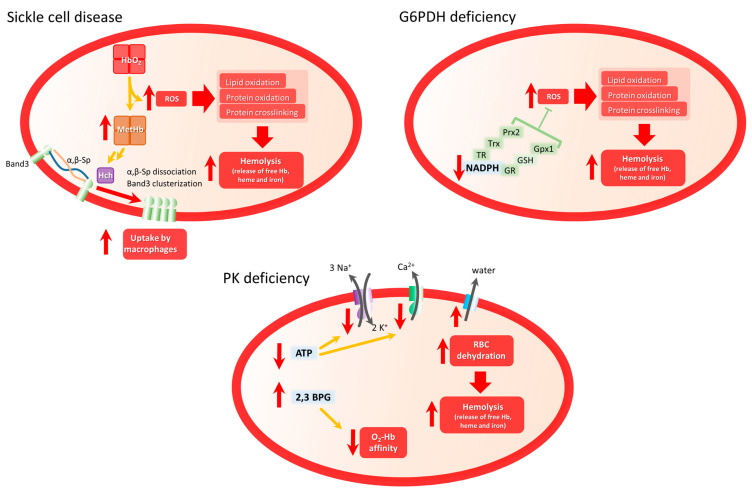
Scheme of main mechanism involved in oxidative stress and hemolytic clinical manifestations in Sickle Cell Disease, G6PDH deficiency, and PK deficiency. In SCD, highly unstable HbS will be converted in MetHb, favoring band 3 clustering and dissociation from membrane complexes, inducing membrane disorganization and membrane fragility. In G6PD, dramatic reduction of NADPH levels diminishes the antioxidant capacity of RBCs increasing ROS-induced hemolysis. In PKD, diminished ATP levels affect the functioning of membrane proteins such as Na^+^/K^+^ pump or PMCA pump, which will indirectly induce water efflux and RBC dehydration, incrementing RBC fragility and hemolysis.

## Data Availability

Not applicable.

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
