# Peer review of "Oxidative Stress in Healthy and Pathological Red Blood Cells"

_biomolecules, 2023, doi:10.3390/biom13081262_

Round 1
Reviewer 1 Report
The authors made a very comprehensive review on the oxidative insults that the RBCs undergo during their life in the circulation and stored in blood bank. Moreover, the authors discussed in detail the three major RBCs pathologies that alters their capabilities to withstand oxidative stresses. The review is well written and understandable. For these reasons I recommend acceptance after minor revisions that are essentially related to one aspect:
in my opinion this review will be improved by the addition of a discussion of some papers from groups that studied oxidative stresses on isolated human RBCs aged under in vitro conditions. Such as high-resolution techniques to directly visualize the oxidative stresses:
Lenzi, E, Dinarelli, S, Longo, G, Girasole, M, & Mussi, V (2021). Multivariate analysis of mean Raman spectra of erythrocytes for a fast analysis of the biochemical signature of ageing. Talanta, 221, 121442
Kozlova, E., Chernysh, A., Sergunova, V., Gudkova, O., Manchenko, E., & Kozlov, A. (2018). Atomic force microscopy study of red blood cell membrane nanostructure during oxidation‐reduction processes. Journal of Molecular Recognition, 31(10), e2724.
Kozlova, E., Chernysh, A., Moroz, V., Kozlov, A., Sergunova, V., Sherstyukova, E., & Gudkova, O. (2021). Two-step process of cytoskeletal structural damage during long-term storage of packed red blood cells. Blood Transfusion, 19(2), 124.Ruggeri, FS, Marcott, C, Dinarelli, S, Longo, G, Girasole, M, Dietler, G, & ... (2018). Identification of oxidative stress in red blood cells with nanoscale chemical resolution by infrared nanospectroscopy. International journal of molecular sciences, 19(9), 2582
Dinarelli, S, Longo, G, Krumova, S, Todinova, S, Danailova, A, Taneva, SG, & ... (2018). Insights into the morphological pattern of erythrocytes' aging: coupling quantitative AFM data to microcalorimetry and Raman spectroscopy. Journal of Molecular Recognition, 31(11), e2732
While two very recent papers were devoted to the discussion of Favism RBCs, and their results were in perfect agreement with what was discussed in the review of the G6PD deficiency.
Girasole, M, Dinarelli, S, & Longo, G (2023). Correlating nanoscale motion and ATP production in healthy and favism erythrocytes: a real-time nanomotion sensor study. Frontiers in Microbiology, 14, 1196764
Dinarelli, S, Longo, G, Germanova-Taneva, S, Todinova, S, Krumova, S, & ... (2022). Surprising structural and functional properties of Favism erythrocytes are linked to special metabolic regulation: a cell aging study. International Journal of Molecular Sciences, 24(1), 637
Author Response
We thank the reviewer for his/her positive comments. In the revised version we have added at the end of third paragraph in page 3 (lines 128-129): ” Atomic force microscopy showed that spectrin filaments are altered during storage, forming thicker fibers and loss of connections (Kozlova et al. Two-step process of cytoskeletal structural damage during long-term storage of packed red blood cells, Blood Transfusion, 19(2), 124).” This was the only paper that used RBCs aged in transfusion bag conditions, and not starvation as the other papers did.
We have also included a paragraph to discuss the possible stress resistance of a stored G6PD RBC subpopulation (page 11, lines 515-522), including related references (Dinarelli et al, Int J Mol Sci, 2022; Francis et al, J Clin Invest, 2020)
Reviewer 2 Report
This is a complete full review of the oxidizing agents that can act on normal red blood cells and in different pathological situations. For each case, the existing defensive mechanisms against oxidative injury is described. The design and content of this review meet the standard criteria for concepts updating and dissemination purpose. I have no comments to add
Author Response
We thank the reviewer for his/her positive comments.
Reviewer 3 Report
The review by Orrico et al concerns an important subject of free radical/oxidation research in biology and medicine. The review is well organized, written in an efficient way for being understood by a large audience.
I suggest minor amendments as follows:
- in section 1.1 page 2, the second paragraph regards PUFA content in RBC which is the main target of oxidation. It should be detailed that PUFA content in RBC is consistently connected with the presence of plasmalogens, as one of the phospholipid classes present in RBC membranes. A specific remark should be done at this point to the analysis of RBC oxidative stress exposure; in fact, according to a recent paper (Biomolecules 2023, 13, 730), analysis of lipid classes must avoid acidic conditions that affect plasmalogen's integrity with an "apparent" loss of PUFA not due to oxidative stress but to the chemical protocols.
- in the References there are minor errors for the correct style.
As far as English is not mother language for this referee, I think the quality of the English Language is fine.
Author Response
We thank the reviewer for his/her positive comments. In the revised version we have added at the end of page 2 (lines 96-99): “... Also, a minor fraction of phospholipids (10-15 %) is present as plasmalogens, which present a vinyl ether hydrocarbon group that has been associated with lipid antioxidant capacity in vitro (Ferreri et al, Biomolecules 2023; Sindelar et al, Free Rad Biol Med, 1999)
We have also corrected the style mistakes in the references.